# The Thickness and Structure of Dip-Coated Polymer Films in the Liquid and Solid States

**DOI:** 10.3390/mi13070982

**Published:** 2022-06-22

**Authors:** Zhao Zhang, Fei Peng, Konstantin G. Kornev

**Affiliations:** Department of Materials Science and Engineering, Clemson University, Clemson, SC 29634, USA; zhaoz@g.clemson.edu (Z.Z.); fpeng@clemson.edu (F.P.)

**Keywords:** dip coating, Landau-Levich-Derjaguin theory, polymer solutions, solid films

## Abstract

Films formed by dip coating brass wires with dilute and semi-dilute solutions of polyvinyl butyral in benzyl alcohol were studied in their liquid and solid states. While dilute and semi-dilute solutions behaved as Maxwell viscoelastic fluids, the thickness of the liquid films followed the Landau-Levich-Derjaguin prediction for Newtonian fluids. At a very slow rate of coating, the film thickness was difficult to evaluate. Therefore, the dynamic contact angle was studied in detail. We discovered that polymer additives preserve the advancing contact angle at its static value while the receding contact angle follows the Cox–Voinov theory. In contrast, the thickness of solid films does not correlate with the Landau-Levich-Derjaguin predictions. Only solutions of high-molecular-weight polymers form smooth solid films. Solutions of low-molecular-weight polymers may form either solid films with an inhomogeneous roughness or solid polymer domains separated by the dry substrate. In technological applications, very dilute polymer solutions of high-molecular-weight polymers can be used to avoid inhomogeneities in solid films. These solutions form smooth solid films, and the film thickness can be controlled by the experimental coating conditions.

## 1. Introduction

Advances in coating processes have evolved in parallel with new technological advances and material development [1,2,3,4] and can be classified with respect to the processing protocols as dry processes (CVD, PVD, PECVD, etc.) [5,6] and wet processes (spray coating, brush casting, spin coating, dip coating, etc.) [7,8,9,10]. Among these coating methods, dip-coating is always attractive because of its simplicity, low cost, and facile control [7,11].

Dip coating of Newtonian viscous fluids on different substrates has been studied in various applications and significant progress in the understanding of the fluid mechanics of film formation has been achieved [11,12,13,14,15,16,17,18,19,20]. Among different substrates, fibers and rods are the most challenging to coat: the curvature of the cylinder significantly influences the film deposition kinetics by altering the flow pattern in the meniscus region [21,22]. The classical Landau-Levich-Derjaguin (LLD) theory of dip coating of fibers (see Equation (10) below) can be used only in a narrow range of fiber radii, when the fiber radius *R* is much smaller than the capillary length $l=\sqrt{\frac{\sigma }{g\rho_{l}}}$, $\frac{R}{l}\;\leq 0.13$, where $\rho_{l}\;$is the density of the liquid, g is the acceleration due to gravity, and σ is the surface tension [23]. This limitation adds to the difficulties of predicting the behavior of coating films formed from complex fluids such as polymer solutions.

In all wet coating processes, including dip coating, one needs to control the final thickness of the coating layer after its solidification [7,8,21,24,25,26,27,28,29,30]. In many cases, the coating liquid is made of rheologically complex fluids [8,29,31,32,33,34], which are difficult to handle and evenly deposit on substrates. For example, almost all ceramic precursors for ceramic coatings, such as Al_2_O_3_, TiO_2_, and mullite precursors, exhibit non-Newtonian features such as viscoelasticity [35] or nonlinear viscosity [36,37,38]. This complexity of the rheological behavior of coating liquids has challenged materials scientists [7,17,22,37,39,40]. Currently, films from polymer solutions demonstrating viscoelastic properties show results that often contradict existing theories [22,41,42,43]. Even after achieving uniformity of the liquid film, it remains a great challenge to convert this liquid film to a uniform and smooth solid layer [4,7,16,29,36,44,45]. At the same time, to satisfy the property requirements for electrical [46,47], thermal [48], and optical [49] films on many devices, one needs to control the thickness and morphology of the remaining solid films. In general, there is no established relation between the thicknesses of the final solid film and the liquid film that was originally deposited on the substrate [11].

To the best of our knowledge, there are no systematic studies of solid polymeric films in relation to their coating prehistory. Only recently, the deficiency in the understanding of the transition from liquid to solid films has been appreciated by materials scientists and new studies have started to address this problem [4,7,16,29,36,44,45].

This paper aimed to systematically analyze the effect of polymer additives on the properties of solid films formed by these polymers. We questioned how the engineering conditions of dip coating, such as the speed of withdrawal of the rods and plates, and the viscosity and surface tension of the coating liquids, influence the thickness and morphology of solid films. To answer these questions, we (1) first studied the effect of the viscoelasticity on the thickness of the liquid film obtained by dip coating of rods and plates; (2) then, we experimentally investigated the relation between the thickness of the liquid and solid films, questioning whether the LLD theory could correctly predict the thickness of the solid polymer film assuming that the evaporating solvent contributes to the film shrinkage proportionally to its volume fraction; and (3) lastly, we studied the effect of the molecular weight of the polymer on the morphology of the dried film.

To clearly observe the effect of the polymer additives on the solution rheology, we used benzyl alcohol (Sigma-Aldrich, 99.8%), a Newtonian solvent that has: (1) sufficiently small viscosity (of the order of the water viscosity); so, the effect of the polymer additives could be easily established; (2) negligible evaporation; so, the mass gain after the dip coating of an article could be easily measured. The high-molecular-weight polyvinyl butyral (PVB, Butvar B-72, Mw = 175,000~250,000 Da) and low-molecular-weight PVB (Butvar B-98, Mw = 40,000 and 70,000 Da), which are both soluble at room temperature in benzyl alcohol, were chosen as polymer additives. These polymers allowed us to study the effects of the polymer elasticity.

## 2. Experimental Methods

### 2.1. Solution Preparation

B72 (PVB, Butvar B-72, Mw = 175,000~250,000 Da) and B98 (PVB, Butvar B-98, Mw = 40,000~70,000 Da) were dissolved in benzyl alcohol and mixed using a magnetic stirrer. A series of solutions with the polymer concentration varying between 0 and 6 g/dL for both B72 and B98 solutions were prepared. Solutions with low concentrations of polymers (0, 0.25, 0.5, and 1 g/dL) were used to investigate the influence of the solution viscoelasticity on the thickness of the coating films in the liquid and solid states. Solutions with higher concentrations of PVB (1~6 g/dL) were also used to investigate the critical overlap concentration of the given polymer solution and the effect of the polymer concentration on the solution viscoelasticity and, finally, on the film thickness.

### 2.2. Rheological Properties

The surface tension of the Newtonian liquids and polymer solutions with different concentrations of PVB in benzyl alcohol was measured using the pendant drop method with the KRUSS DSA10 instrument with a 0.5 mm needle at room temperature. Images of the pendant drops were captured and analyzed using KRUSS DSA software.

The density of the fluids was calculated by measuring the weight of the fluid in a fixed-volume liquid pycnometer (25 mL) using an analytical balance (PI-214, Denver Instrument) at room temperature.

The viscosity of the Newtonian liquids was measured using a viscometer (Brookfield DV3THBCJ0). The viscosity of the polymer solution was measured using the capillary method. UBBELOHDE-type viscometers were kept in a water bath (Cannon CT-1000) at 25 °C and a stopwatch was used to measure the efflux time. Here, the efflux time was maintained above 100 s in order to minimize the experimental error.

The critical overlap concentration $c^{*}$ of the PVB-benzyl alcohol system was determined by the single-point and multipoint methods.

To calculate the overlap concentration $c^{*}$ using the single-point method, the viscosity of the pure benzyl alcohol and 1 g/dL PVB-B72 solution was measured. The intrinsic viscosity of PVB-B72 and the overlap concentration in benzyl alcohol were determined as [50]:(1)$$\left[\eta \right]=\frac{\sqrt{2\left(\eta_{rel}-1\right)-2\ln\eta_{rel}}}{c},\;\;\;c^{*}=\frac{1}{\left[\eta \right]}$$
where $\eta_{rel}=\frac{\eta }{\eta_{0}}$ is the relative viscosity; η is the viscosity of the polymer solution; *c* is the polymer concentration; and $\eta_{0}$ is the viscosity of the pure solvent.

To determine the overlap concentration using the multipoint method, the viscosity vs. PVB concentration was plotted on a log-log scale. Then, the two power law fits were calculated for the data points on either side of the apparent slope change; at least 3 data points near this region were used for each fit. To determine the intersection of these fits, they were set equal, and the multipoint overlap concentration was calculated as the intersection point.

The viscoelasticity of a fluid can be characterized by its mean relaxation time. Figure 1a shows a schematic diagram of a filament break-up experiment; using this diagram, one can obtain the relaxation time by observing how the mid-diameter $D_{mid}$ of a filament decreases with time [51,52,53,54]. For Newtonian fluids with no elasticity, the filament will neck down and break up rapidly due to the Rayleigh–Tomotika instability and $D_{mid}$ will decrease linearly with time [55,56]. However, for fluids with elasticity, this process will take a longer time, and $D_{mid}$ should be proportional to $\exp\left(-\frac{t}{3\lambda_{1}}\right)\;$[51,52,53,54,57], where $\lambda_{1}$ is the relaxation time.

Using a homemade filament stretching device, a small quantity of fluid was first placed between two plates and then stretched by moving two needles apart very quickly. After stretching, a filament formed between the ends of the two plates, with its diameter $D_{mid}$ decreasing with time. This process was recorded using high-speed cameras (MotionPro X3 and MotionPro Y) with a frame rate of up to 5000 frames per second.

### 2.3. Measurements of the Thickness of Liquid Films

Brass wire with a radius of 0.255 mm was chosen as the coating substrate. Before dip coating, all the wires were sonicated by an ultrasonicate cleaner (VWR Ultrasonic Cleaner 97043-964, 35 kHz, 90 W) in acetone for 2 h and dried thoroughly.

The dip coater (KSV NIMA Dip Coater, Biolin Scientific Oy, Espoo, Finland) was used to make these coatings. During the dip coating process, the wire was attached to the dip coater by a clamp. It was ensured that the wire was fixed vertically. The wire was lowered into the fluid reservoir at a velocity of 100 mm/min. After reaching the targeted length, the wire was kept still for 15 s before withdrawal from the reservoir. Five withdrawal velocities (U=20, U= 50, U= 100, U= 200, and U= 500 mm/min) were chosen to study the dependence of the coating thickness on the capillary number Ca=Uη/σ.

The meniscus shape during the dip coating process was recorded using a high-resolution camera (Grasshopper 3, FILR, Wilsonville, OR, USA); an example of the meniscus shape is shown in Figure 2a. It is shown that the meniscus videos have sufficient resolution to provide information on the shape of the meniscus close to the wire surface [58,59].

The height and focus of the camera were adjusted so that the horizontal air–liquid interface could be captured, and the wire was sufficiently magnified. The light source intensity and angle were carefully adjusted to make sure the meniscus shape and brass wire could be easily differentiated from the surroundings.

The parameters of the images, such as the hue and brightness/contract, were adjusted using Photoshop to ensure that the profile of the meniscus boundary was clear and easily identified (Figure 2a). These images were subsequently converted to black and white (Figure 2b) and their profile analyzed using the theoretical meniscus profile [60]:(2)$$z(\frac{R+h}{R})=\;-\mathrm{Rcos}\theta \left[ln\left(\left(\frac{R+h}{R}\right)+\sqrt{{\left(\frac{R+h}{R}\right)}^{2}-{\left(\cos\theta \right)}^{2}}\right)+ln\left(\frac{e^{E}\sqrt{\beta }}{4}\right)\right]\;+\mathrm{Rcos}\theta \left[K_{0}\left(\left(\frac{R+h}{R}\right)\sqrt{\beta }\right)+ln\left(\frac{R+h}{R}\right)+ln\left(\frac{e^{E}\sqrt{\beta }}{2}\right)\right]$$

In this equation, z((R+h)/R) is the height of the meniscus profile at a certain radius (R+h) measured from the wire axis, θ is the contact angle, *E* = 0.577215 is the Euler constant, $\beta =\frac{\left(\rho_{l}-\rho_{g}\right)gR^{2}}{\sigma }$ is the Bond number, and
$K_{0}\left(\frac{R+h}{R}\right)$ is the Bessel function of the second kind. A Matlab code was developed to find the dynamic contact angle by adjusting the horizon line. Each side of the meniscus was examined, and the dynamic contact angle was reported as an average of these two. In Figure 2b, we illustrate this procedure: the contact angle for the left side of the meniscus was found to be θ= 15° while on the right side, it was θ= 18°. Thus, we report θ=(15°+18°)/2=16.5°. The error bar represents the standard deviation of the contact angles measured in 3 repeats.

### 2.4. Measurements of the Thickness of the Dried Films and Their Morphology

To study the morphology of the dry films, we used an atomic force microscope (AFM, Alpha300, Witec Instruments Corp., Ulm, Germany). Since the curvature of brass wire is high, it is difficult to scan its surface with AFM. Therefore, we prepared flat substrates for this purpose. Silicon wafers (WRS Materials) were used as substrates. The Si wafers were first cut into 10 mm × 50 mm pieces using a diamond cutter. They were cleaned in an ultrasonic bath (VWR Symphony, Suwanee, GA, USA) for 30 min with deionized water. Subsequently, the wafers were placed in a hot (~60 °C) “piranha” solution (3:1 concentrated sulfuric acid/30% hydrogen peroxide) for 1 h of sonication. Then, the wafers were rinsed several times with high-purity deionized water and stored in deionized water. Before use, the substrates were dried under a stream of dry high-purity nitrogen (National Specialty Gases, Greenville, SC, USA).

To confirm that the Si wafers had the same wetting properties as our wires, using Krüss, FM40MK2 Easydrop, we measured the contact angles that the sessile drops of PVB solutions formed with them. The solutions with the lowest and highest concentrations (0.25 g/dL PVB-B72 and 1 g/dL PVB-B72) were used for the comparison of their contact angles. These solutions have very similar surface tensions (Table 1 and Table 2). The advancing contact angles of 0.25 g/dL PVB-B72 and 1 g/dL PVB-B72 on Si wafers were 21.0° ± 0.32° and 21.1° ± 0.49°, respectively, which are very similar to the advancing contact angle of PVB solution on brass wire (~22°). Thus, the solutions with intermediate concentrations of polymers are expected to show similar contact angles. These results confirmed that the adhesion characteristics of Si wafers should be similar to those of brass wires and, hence, the AFM results should provide useful insight into the morphology of the films obtained on brass wires.

The dip coater (KSV NIMA Dip Coater, Biolin Scientific Oy, Espoo, Finland) was used to reproduce coatings on the Si wafer. During the dip coating process, the Si wafer was attached to the dip coater by a clamp. It was ensured that the wafer was fixed vertically. The coating procedure explained in the previous section was strictly followed.

The coated substrates were dried vertically in the oven at 80 °C overnight.

To measure the coating thickness after drying, we scratched the solid film along the vertical line in the center of the substrate. The edges of the scratched trenches were scanned using an atomic force microscope (AFM, Alpha300, WITec Instruments Corp.) to obtain the difference in height between the center of the surface of the PVB coating and the substrate. The thickness of the coating was measured 15 mm from the bottom of the substrate. The thickness profile along the vertical dimension was obtained by measuring the thickness from position 2.5 mm to position 27.5 mm with a step of 2.5 mm.

The coating thickness and roughness of the coating were evaluated using Gwyddion open-source software. To calculate the roughness, 5 lines were drawn on the AFM image. The root mean square roughness (*R_q_*) was then evaluated by the software following its definition:(3)$$R_{q}=\sqrt{\frac{1}{n}\overset{n}{\underset{i=1}{\sum }}y_{i}^{2}}$$

Due to the film drainage [61] and the edge effects during the drying process, the coating thickness of the dried film follows a parabolic profile along the vertical coordinate. This profile follows the Jeffreys expression for post-withdrawal drainage of a Newtonian fluid [62]:(4)$$h\left(z,t\right)={\left(\frac{\eta z}{\rho gt}\right)}^{1/2}$$

If we neglect the end droplet and assume that the profile of a liquid film is parabolic, the average coating thickness at any time t is obtained as:(5)$$\bar{h}\left(t\right)=\frac{1}{L}\int_{0}^{L}{\left(\frac{\eta z}{\rho gt}\right)}^{1/2}dz=\frac{2}{3}{\left(\frac{\eta }{\rho gt}\right)}^{1/2}L^{1/2}$$
where L is the coating length. Solving Equation (5) for t and plugging the result into Equation (4), we obtain the position $z_{\bar{h}}\;$as:(6)$$z_{\bar{h}}=\frac{4}{9}L$$

This position specifies the place where the average coating thickness can be identified (Figure 3). In our experiment, the coating length was L=35 mm, which gives $z_{\bar{h}}\approx 15.5\;\mathrm{mm}$. Thus, our selection of the measurement position (15 mm) reflects the “dry” coating thickness of the sample:(7)$${\bar{h}}_{dry}\approx h{\left(15\mathrm{mm}\right)}_{dry}$$

The theoretical “dry” coating $h_{\infty dry}$ thickness was calculated as:(8)$$h_{\infty dry}=\frac{V_{P}h_{\infty p}}{V_{P}+V_{S}}=\frac{\frac{m_{P}}{\rho_{P}}h_{\infty p}}{\frac{m_{P}}{\rho_{P}}+V_{S}}=\frac{c_{P}h_{\infty p}}{c_{P}+\rho_{P}}$$

Where $c_{P}\;$is the polymer concentration in solvent ($c_{P}=\frac{m_{P}}{V_{S}}$); $\rho_{P}$ is the density of the polymer; and $h_{\infty p}$ is the coating thickness based on the LLD theory for plates, given as:(9)$$h_{\infty p}=A_{plate}\;\cdot \;l\;\cdot \;{\mathrm{Ca}}^{\frac{2}{3}}\;,\;\;\;\;\;A_{plate}=0.945,\;\;\;\;\mathrm{Ca}=U\;\eta /\sigma ,\;l=\sqrt{\frac{\sigma }{g\rho_{l}}}$$

## 3. Results and Discussion

### 3.1. Density, Surface Tension, and Viscosity

Table 1 and Table 2 shows the surface tension, density, and calculated capillary length of PVB solutions with various concentrations. Table 3 and Table 4 report the solution viscosity.

Based on these data, we conclude that the addition of small quantities of PVB to benzyl alcohol does not cause much change in the surface tension, density, or the capillary length of the solution. Plotting the viscosity as a function of the concentration in Figure 4a, it is observed that the slope of the viscosity vs. PVB-B72 concentration changes suddenly between 0.5 and 0.6 g/dL, where the critical overlap concentration is located. After extending the regression lines below and above the critical overlap concentration, the precise value of $c^{*}$ was determined to be 0.52 g/dL. The same method was applied to determine the critical overlap concentration of PVB-B98 (1.83 g/dL), as shown in Figure 4b.

Using the single-point method [50], we found that the intrinsic viscosity of the PVB-B72-benzyl alcohol system is 1.806 dL/g. Based on the relationship between the critical overlap concentration $c^{*}$ and intrinsic viscosity [η], we calculated [η]=0.56 g/dL for B72, which is consistent with the multipoint method.

### 3.2. Relaxation Time

The filament breakup process was monitored using a high-speed camera (Motion Pro X3, Princeton Instruments, Trenton, NJ, USA). An example (PVB-B72, 2 g/dL) of how the filament mid-diameter changes with time is shown in Figure 5a. These images were then analyzed with our Matlab code to determine the filament mid-diameter. The relaxation time $\lambda_{1}$ was determined by fitting $D_{mid}$ with $\exp\left(-\frac{t}{3\lambda_{1}}\right)$ [57]. Figure 5 shows the linear change in the relaxation time with the concentration of PVB. Considering the measured $c^{*}$ = 0.52 g/dL (PVB-B72) and $c^{*}$ = 1.83 g/dL (PVB-B98), it is clearly seen that the Maxwell model with a single relaxation time can describe the solutions far above the critical overlap concentration.

### 3.3. Effect of Solution Viscoelasticity on the Coating Thickness

Our results in Figure 6 show that the film thickness $h_{\infty f}\;$extracted from the measurements of flux *q*, as discussed in detail in our recent publication [23], follows the LLD theory for fibers ($h_{\infty f}$) when $Ca>10^{-4}$:(10)$$h_{\infty f}=A_{fiber}\;\cdot \;R\;\cdot \;{\mathrm{Ca}}^{\frac{2}{3}},\;\;A_{fiber}=1.34$$

No discrepancy with the Newtonian behavior was found when we plotted the film thickness versus capillary number, as shown in Figure 6. Neither high-molecular-weight PVB-B72 nor low-molecular-weight PVB-B98 show any significant deviation from the LLD theory. The data obtained on lower capillary numbers ($\mathrm{Ca}<10^{-4}$) must be interpreted with caution as the films were very thin and hence, an incremental change in the weight of the brass wires was difficult to measure accurately because of the limitation of our semi-micro balance.

To explain the observed trend, we compared the characteristic time associated with the shearing of liquid in the film and the time associated with the extension of polymers in the shear flow. The shear rate in the dip-coating process can provide a reasonable estimate of the characteristic time for the polymer extension:(11)$$t_{viscous}=\frac{h}{U}$$

Taking $h=h_{\infty f}\;\sim 20\;\mu m$ and at the maximum withdrawal velocity U=500 mm/min, we have $t_{viscous}\approx 2.4\times 10^{-3}$ s. The relaxation time of our polymer solutions is less than one millisecond, $\lambda_{1}<10^{-3}\;s$. Therefore, the ratio $\frac{\lambda_{1}}{t_{viscous}}\;1.$ This small ratio indicates that during the coating process, the polymer has time to relax. Hence, any elastic effects of polymer coils can be safely ignored in this range of speed of dip coating.

### 3.4. Dynamic Contact Angle

Figure 7 show the effect of the polymer concentrations and velocities on the advancing and receding dynamic contact angles for both the high-molecular-weight PVB and the low-molecular-weight PVB. For the advancing dynamic contact angle, there is no clearly observable differences between the molecular weights. For the receding contact angle, the higher the molecular weight of the polymer, the stronger its velocity dependence. Thus, at the same polymer concentration, the greater the solution viscosity, the smaller the receding contact angle.

As shown in Figure 7, the continuum fluid mechanics theory does explain the relations between the coating thickness and capillary number. We, therefore, applied the same type of continuum fluid mechanics model, the Cox–Voinov model [63] of the dynamic contact angle, to understand the obtained trends in Figure 8. In the fluid mechanics theory, the apparent dynamic contact angle $\theta_{dyn}$ is predicted by the Cox–Voinov model as:(12)$$\theta_{dyn}^{3}=\theta_{0}^{3}\pm ACa$$
where *A* is a positive constant and $\theta_{0}$ is the static advancing contact angle when the substrate is not moving. The positive sign in Equation (12) refers to the advancing meniscus and the negative sign refers to the receding meniscus. The angle $\theta_{0}$ is obtained by setting the velocity of the moving contact line to zero U=0, i.e., *Ca* = 0. Equation (12) suggests that the cubed dynamic angle should linearly depend on the velocity of the moving wire and the slope of this dependence should not depend on the direction of wire movement. In Figure 8, the experimental dynamic contact angles versus log Ca are plotted to cover the entire range of capillary numbers [64,65].

In contrast to the Cox–Voinov prediction, in the range of capillary numbers, 10^−6^ < Ca < 10^−4^ where the theory should work, the advancing contact angle (Figure 8a) does not depend on the capillary number. The receding contact angle (Figure 8b) does depend on the capillary number, and it does decrease linearly as the capillary number increases. Thus, within the studied range of capillary numbers, one would not see any elastic contribution of polymers when the wire is withdrawn from polymer solutions. We did not observe any behavior similar to behavior reported in the literature [66] that has been associated with the solvent evaporation and polymer concentration at the receding contact line. In contrast, dipping the wire into solution, one would observe some deviation from the Newtonian behavior, suggesting that the polymer interacts more strongly within the liquid wedge in this mode of movement and the fluid flow is screened by the polymer chains, preserving the equilibrium shape of the meniscus.

### 3.5. Coating Thickness of Solid vs. Liquid Films

Figure 9a is an example of the edge profile of the scratch trench taken by AFM and analyzed by Gwyddion. The thickness of the solid film was measured by calculating the height difference at the edge of the step. The thickness along the vertical dimension follows a parabolic profile (Figure 9b). The only exceptions are given by the measurements taken at the points that are very close to the bottom edge (position = 2.5 mm). At this position, the drained fluid carrying the polymer gathered into a big drop before the solvent had completely evaporated.

As shown in Figure 9b, the film thickness at the heights between 10 and 18 mm did not change significantly. Therefore, we analyzed the dependence of the thickness of the dry film at the height 15 mm, $h\;{\left(15\;\mathrm{mm}\right)}_{dry}$. With the increasing capillary number, the ratio $\frac{h\;{\left(15\;\mathrm{mm}\right)}_{dry}}{h_{\infty dry}}\;$ decreased (Figure 9c,d), suggesting that the theory overestimates the thickness of the dry film. Since the surface tensions of low- and high-molecular-weight polymers are the same, the main contribution to the capillary number comes from the viscosity and speed of the plate withdrawal. In Figure 9d, we attempt to separate these two contributions, replotting the data as a bar chart. We observe that the theoretical prediction $\frac{h\;{\left(15\;\mathrm{mm}\right)}_{dry}}{h_{\infty dry}}\sim \;1\;$for low-molecular-weight solutions only works for very low speeds of plate withdrawal, 20 mm/min, and for low polymer concentrations < 1 g/dL. At greater withdrawal speeds, the high-molecular-weight solutions perform better. However, the effect of the polymer concentration becomes important: in the range of withdrawal speeds from 50 to 200 mm/min, only polymer solutions with concentrations near the critical overlap concentration (~0.5 g/dL) follow the theoretical prediction. As the withdrawal speed decreases, the dilute solutions make the film thickness thicker than expected.

These data must be interpreted with some caution. Indeed, as shown by Equation (4), the coating thickness at position z will decrease with time due to the film drainage. At a greater capillary number, i.e., at a greater withdrawal speed or smaller viscosity, the thickness of the coating film formed during the dip-coating process should be thicker. Therefore, it took a longer time for the thicker film to dry and more polymer gathered at the bottom edge of the plate. The drainage effect on the coating thickness is especially significant because the solvent in our system (benzyl alcohol) has a very low evaporation rate.

### 3.6. Coating Thickness vs. Coating Roughness

For high-molecular-weight PVB-B72, the thickness of the dried films varied from 13 to 173 nm. For all these samples, the roughness of the coating surface was very low ($R_{q}<1\;\mathrm{nm}$) (Figure 10a or Figure 11a).

However, for the low-molecular-weight PVB-B98 polymer, the roughness of the solid coating showed a thickness dependence (Figure 10b and Figure 11b). For coatings thicker than 40 nm, the surface roughness was low ($R_{q}<1\;\mathrm{nm}$). As the coating thickness decreased, the surface roughness rose quickly (~7 nm). The higher roughness is related to the formation of nodules and nodule aggregates, as shown by the AFM scans (Figure 11b). These nodules and nodule aggregates are most likely formed due to the convection cells caused by surface tension gradients [67]. Scriven et al. [68] found that for thinner films, the surface-tension-driven convection is initiated by smaller temperature differences. On the other hand, the viscosity can resist convective flow [67] and stabilize the coating.

The critical overlap concentrations for PVB-B72 and PVB-B98 were $c^{*}=0.52\;$and $c^{*}=$1.83 g/dL, respectively. Therefore, all polymer solutions used here (except for PVB-B72 1 g/dL) were classified as being dilute or semi-dilute polymer solutions. The viscosity of the solution was also small. During solvent evaporation, the concentration of the polymer in the film increases. In the solutions of the high-molecular-weight polymer (PVB-B72), the viscosity increased rapidly with the concentration (Figure 4a). Therefore, as evaporation proceeds, the rapidly increased viscosity resists the development of film instability. As a result, a smooth solid film remains deposited on the substrate (Figure 10a or Figure 11a).

In contrast, in the solutions of the low-molecular-weight polymer (PVB-B98), the viscosity increased more slowly with the concentration (Figure 4b). In very thin films, less than a 40 nm thickness, this effect of the low-molecular-weight polymers becomes critical in shaping the solid films. In these films, polymer coils do not have time to concentrate: the evaporation of the solvent is fast, and the dried islands prevent the formation of entanglements between the polymers. Due to the slow increase in the viscosity and the lack of polymer entanglements in dilute and semi-dilute polymer solutions, any perturbations of the air–liquid interface during film evaporation were not effectively damped. As a result, nodules and nodular aggregates formed in the thin films, as shown in Figure 10b and Figure 11b. The film’s roughness increased.

When the coating thickness of the solutions of the low-molecular-weight polymer (PVB-B98) increased above >40 nm, the resulting solid film was smooth again. This suggests that over time, the solvent from dilute or semi-dilute polymer solutions evaporates, gradually increasing the solution concentration and hence polymer entanglements. The rapid increase in the viscosity and significant entanglements of polymers effectively resist the development of perturbations at the air–liquid interface, leading to a smoother solid film.

## 4. Conclusions

We studied the effects of polymer additives on the thickness of coated films in their liquid and solid states. The films were formed by dip coating of brass wires and Si wafers. As the coating liquids, we used solutions of polyvinyl butyral in benzyl alcohol. To evaluate the effect of the polymer elasticity and viscosity, solutions with different polymer concentrations were studied. Dilute and semi-dilute polymer solutions were investigated. The critical overlap concentrations of high-molecular-weight polymer PVB-B72 and low-molecular-weight polymer PVB-B98 were determined to be 0.52 and 1.83 g/dL, respectively. The viscoelasticity of the polymer solutions was evaluated using the liquid bridge breakup method. Both polymer solutions demonstrated the features typical of a viscoelastic Maxwell fluid with a single relaxation time.

The coating thickness was studied for a broad range of capillary numbers, 10^−6^ < Ca=Uη/σ < 10^−2^, specifying the effect of the speed U of withdrawal of the article, viscosity *η* of the coating film, and its surface tension, σ. It was found that the thickness of the liquid films of the dilute and semi-dilute PVB polymer solutions was fully described by the Landau-Levich-Derjaguin theory. Thus, polymer viscoelasticity does not influence the thickness of the liquid films in the range of capillary numbers less than Ca < 10^−2^.

By studying in detail the behavior of the dynamic contact angles of the meniscus, we observed that in the range of capillary numbers 10^−6^ < Ca < 10^−4^, the advancing contact angle does not follow the predictions of the Cox–Voinov theory formulated for Newtonian fluids. In this range of capillary numbers, the advancing contact angle remained constant, suggesting an important contribution of the polymer elasticity that prevented any flow from occurring in the liquid wedge adjacent to the contact line. The receding contact angle follows the Cox–Voinov predictions in accordance with the success of the Landau-Levich-Derjaguin theory.

It was found that the coating films of the slowly evaporating PVB solutions experienced the effect of draining during the post-withdrawal evaporation. To study the film thickness in the region where the solid film has almost a constant thickness, we applied the mixture rule for the thickness of the liquid film and derived Equation (8), relating the thickness of the dried film to the thickness of the liquid film. Using Equation (8), we showed that the thickness of the dried film significantly deviates from the LLD theory prediction: Therefore, the estimates of the thickness of the solid residual film based on the knowledge of the liquid film thickness cannot be justified.

To study the film surface morphology, we employed atomic force microscopy and prepared solid films on Si-wafers using the same dip coating method. The Si wafers had the same wetting properties as those of the brass wires. For the high-molecular-weight polymer coatings (PVB-B72), the dried films had smooth surfaces even at a very low coating thickness (<10 nm). Smooth solid films also formed when solutions with the low-molecular-weight polymer (PVB-B98) were used. However, this smooth film was obtained only when the coated films were thicker than 40 nm. In the very thin solid films (<40 nm) obtained from the low-molecular-weight polymer (PVB-B98), nodules and nodular aggregates were observed, and the film surface was rough.

These results suggest that in a broad range of processing parameters, 10^−6^ < Ca=Uη/σ < 10^−2^, smooth solid films with a thickness less than hundreds of nanometers can be formed from dilute polymer solutions. The higher the polymer molecular weight, the thinner the film formed. The conditions for the formation of uniform polymer films thinner than 40 nm remain unknown and further investigations are needed to reveal the possibility of forming nanometer-thick films from low-molecular-weight polymers.

## Figures and Tables

**Figure 1 micromachines-13-00982-f001:**
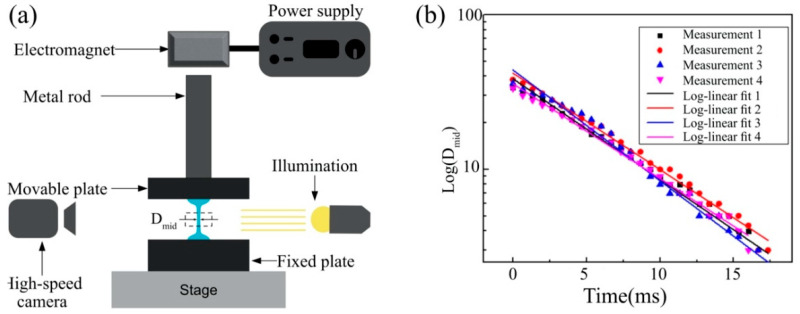
(**a**) Schematic of the filament break-up experiment. (**b**) An example of the determination of the relaxation time $\lambda_{1}$ based on the log-linear fit, log(D_mid_) ∝(−t/(3λ_1_)), where PVB B-72 in benzyl alcohol with a concentration of 4 g/dL was used. Four measurements of the same solution on the same set-up were performed to find an average relaxation time.

**Figure 2 micromachines-13-00982-f002:**
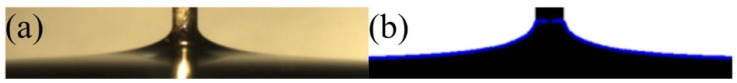
(**a**) An example of an image of a receding meniscus (1 g/dL polyvinyl butyral, 100 µm/s); (**b**) The best fit of the meniscus profile by Equation (2) ((**left part**): 15°, (**right part**) 18°).

**Figure 3 micromachines-13-00982-f003:**
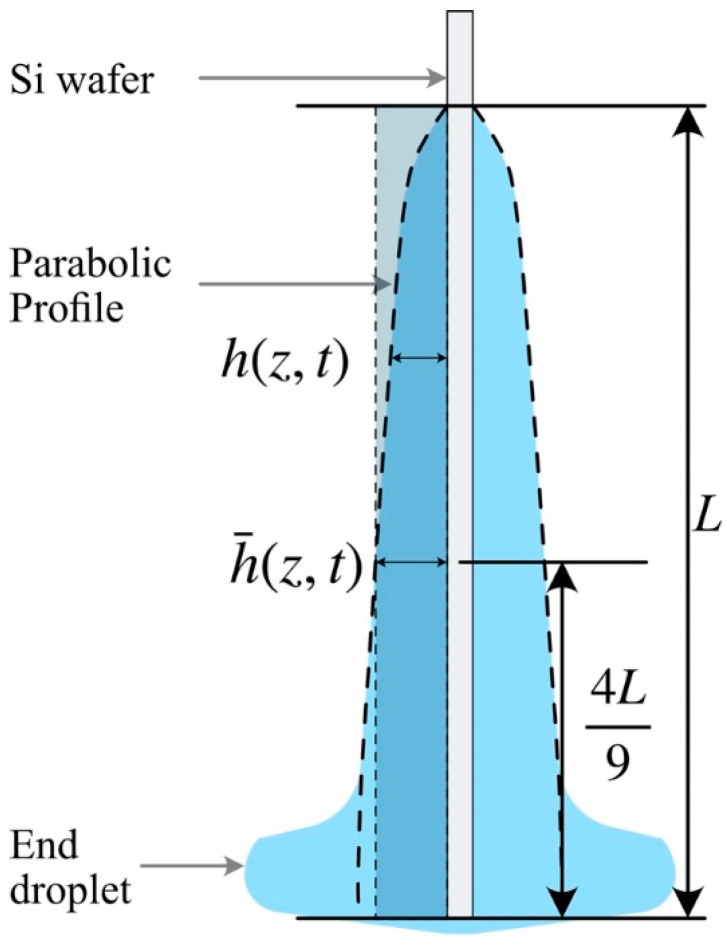
Diagram of the parabolic profile of the coating thickness after film draining at time *t*.

**Figure 4 micromachines-13-00982-f004:**
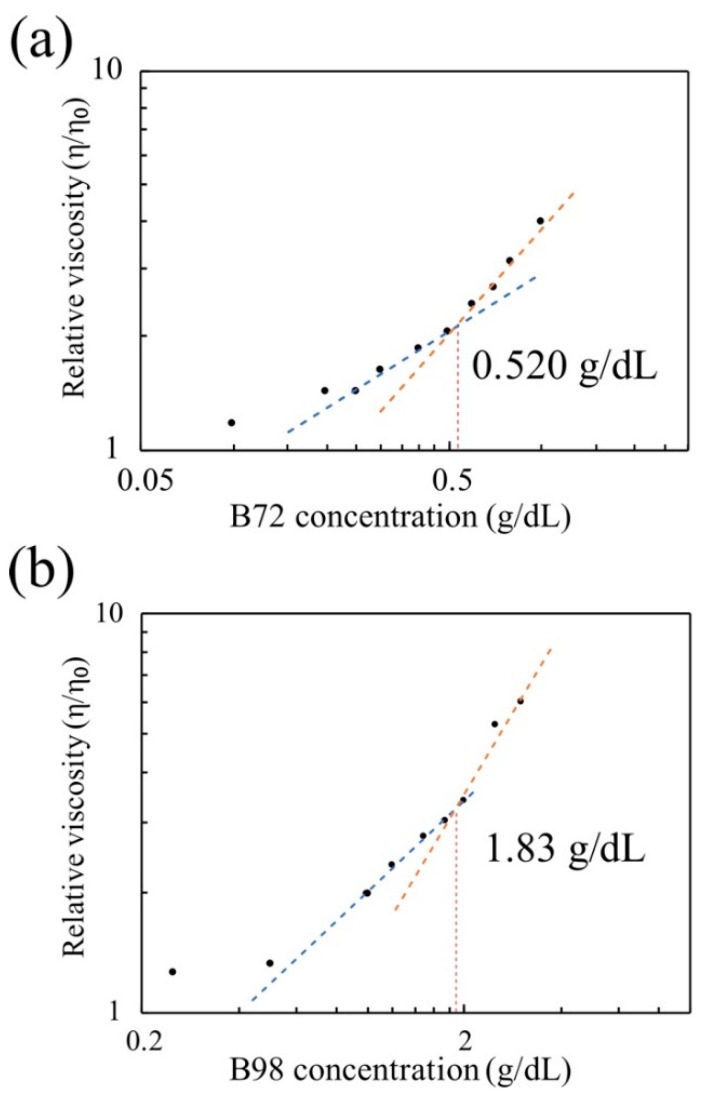
(**a**) Relative viscosity vs. PVB concentration in benzyl alcohol for high-molecular-weight PVB-B72. (**b**) Relative viscosity vs. PVB concentration in benzyl alcohol for low-molecular-weight PVB-B98.

**Figure 5 micromachines-13-00982-f005:**
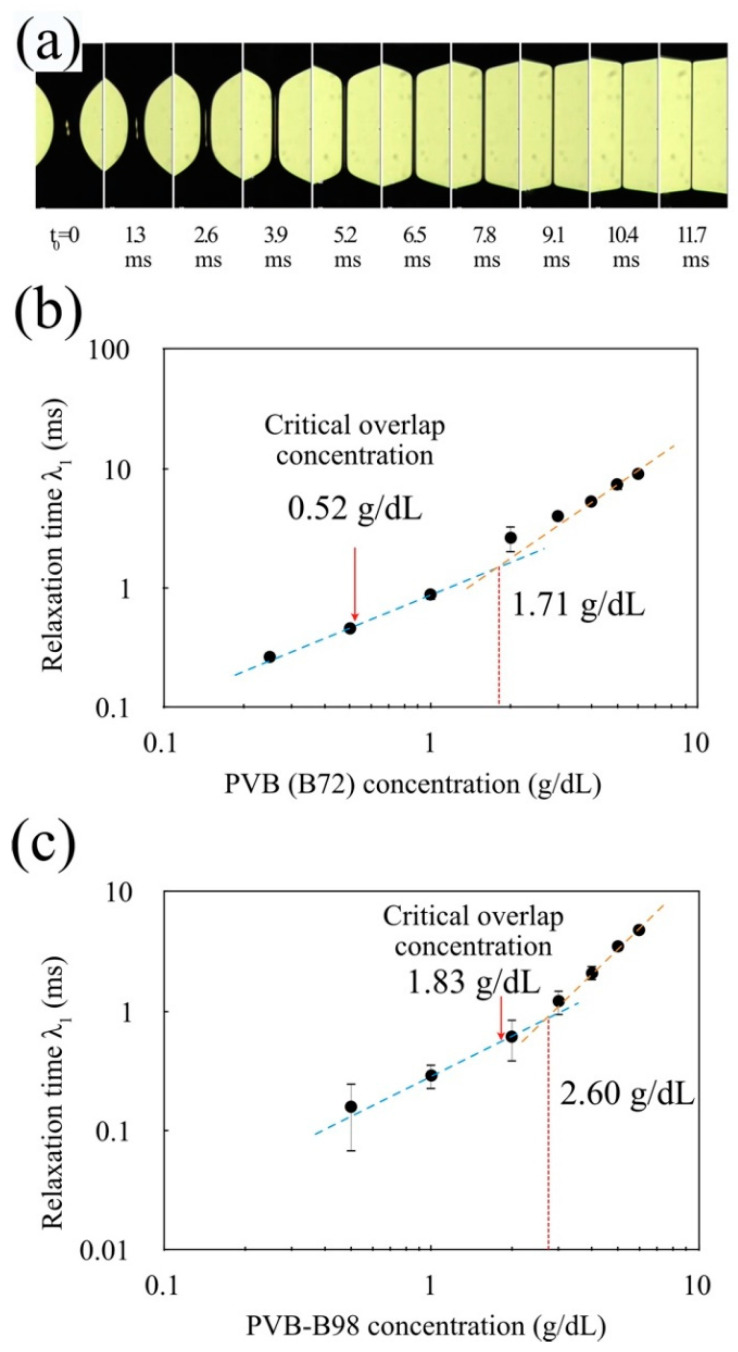
(**a**) Filament break-up experiment with PVB-B72 (2 g/dL) in a benzyl alcohol solution. (**b**) Concentration dependence of the relaxation time for high-molecular-weight PVB-B72. (**c**) Concentration dependence of the relaxation time for low-molecular-weight PVB-B98.

**Figure 6 micromachines-13-00982-f006:**
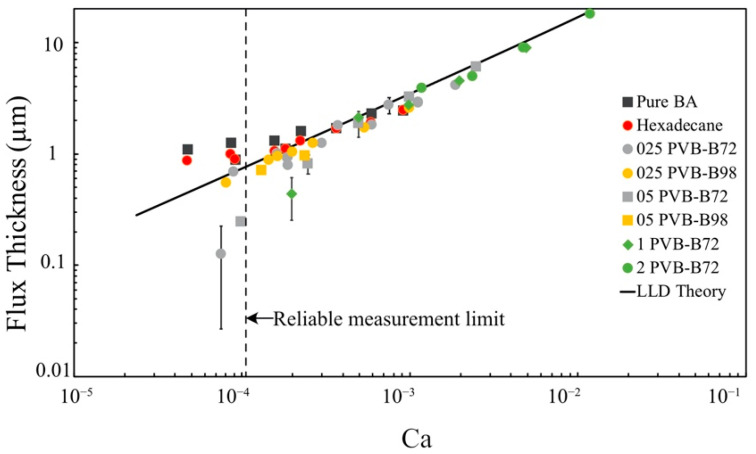
The film thickness on a brass wire (R = 0.255 mm) vs. capillary number. Pure viscous fluids (pure BA, hexadecane) and viscoelastic fluids (PVB-BA solutions) were examined. The solid line is Equation (10), the LLD theory for fibers ($h_{\infty f}$ ). The data on the right side of the dashed vertical line have the greatest accuracy with the given methodology.

**Figure 7 micromachines-13-00982-f007:**
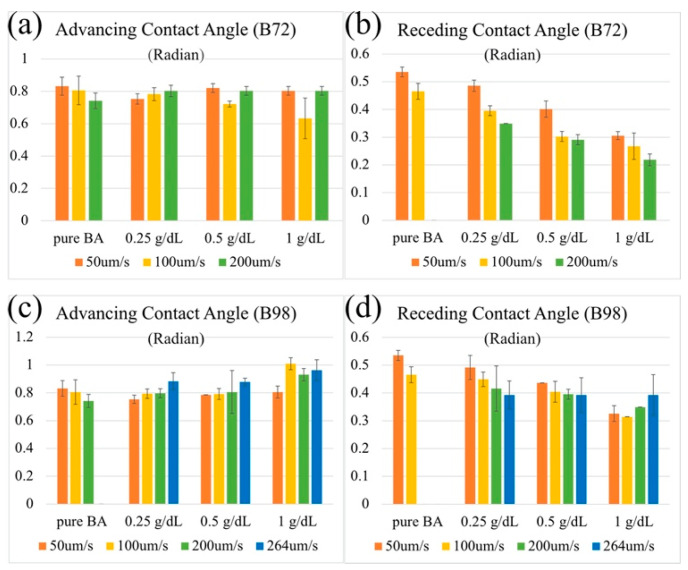
The resulting advancing (**a**,**c**) and receding (**b**,**d**) dynamic contact angles of PVB solutions at different concentrations, velocities, and molecular weights of PVB.

**Figure 8 micromachines-13-00982-f008:**
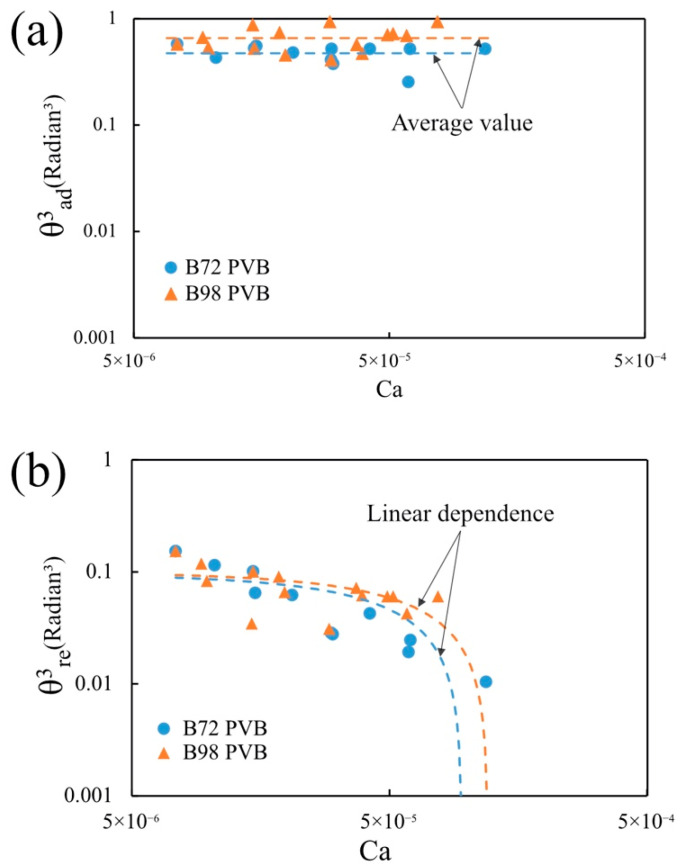
The capillary number dependence in the logarithmic scale of (**a**) the advancing dynamic contact angle $\theta_{a}$ and (**b**) the receding dynamic contact angle $\theta_{r}$. To guide the reader, the dashed curves corresponding to the linear dependences are shown.

**Figure 9 micromachines-13-00982-f009:**
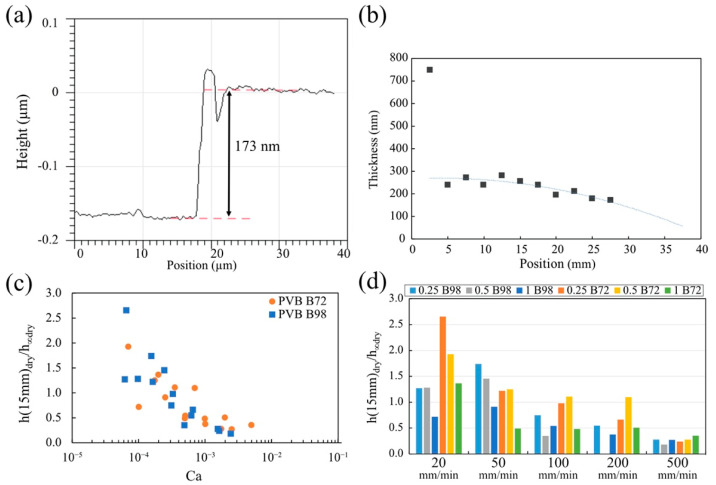
(**a**) The AFM edge profile of the scratched trench. The coating was prepared with 1 g/dL PVB-B72 at a withdrawal velocity of 500 mm/min. (**b**) The profile of the solid film along the vertical direction of the coated plate (position = 0 is at the bottom; position = 35 mm is at the top edge of the coating film). (**c**) The ratio of the measured thickness of the solid film at position 15 mm (*h*(15 mm)*_dry_*) and the thickness based on LLD theory *h__∞dry_* (defined in Equations (8) and (9) vs. the capillary number. (**d**) The ratio of the measured thickness of the solid film at position 15 mm (*h*(15 mm)*_dry_*) over the “dry” coating thickness based on LLD theory h_∞dry_ (defined in Equations (8) and (9)) for different polymer concentrations and withdrawal velocities.

**Figure 10 micromachines-13-00982-f010:**
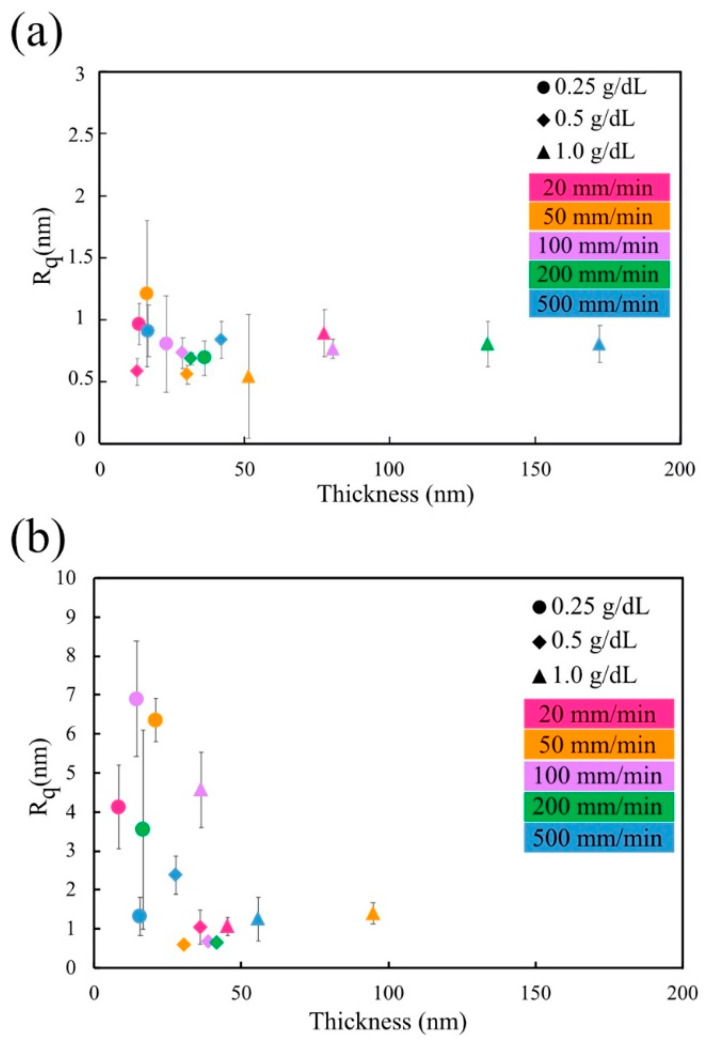
The roughness dependence of the solid film on its thickness: (**a**) high-molecular-weight PVB(B72) solution; (**b**) low-molecular-weight PVB(B98) solution.

**Figure 11 micromachines-13-00982-f011:**
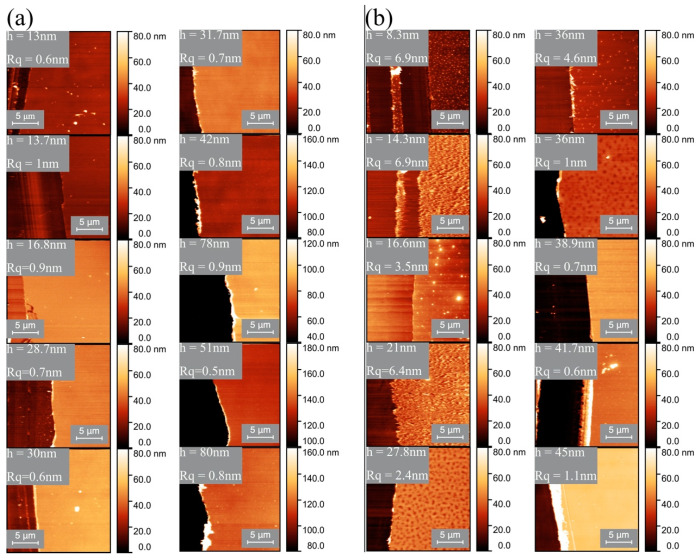
(**a**) AFM images showing the morphology of the solid films obtained from the high-molecular-weight polymer (B72). The films with different thicknesses have smooth surfaces. (**b**) AFM images showing the morphology of solid films obtained from the low-molecular-weight polymer (B98). Thin films have bumpy surfaces or form nodules. In each panel, the bright right part is the film, and the dark left portion is the trench.

**Table 1 micromachines-13-00982-t001:** Density of B72-benzyl alcohol solutions.

B72 Concentration (g/dL)	0	0.25	0.5	1
Density (g/cm^3^)	1.04	1.04	1.04	1.04
Surface tension (mN/m)	37.5	37.3	37.7	37.6
Capillary length (mm)	1.918	1.913	1.923	1.920

**Table 2 micromachines-13-00982-t002:** Density of B98-benzyl alcohol solutions.

B98 Concentration (g/dL)	0	0.25	0.5	1
Density (g/cm^3^)	1.04	1.04	1.04	1.04
Surface tension (mN/m)	37.5	37.5	37.4	37.6
Capillary length (mm)	1.918	1.918	1.915	1.920

**Table 3 micromachines-13-00982-t003:** Viscosity of B72-benzyl alcohol solutions.

[B-72] (g/dL)	0	0.1	0.2	0.3	0.4	0.5	0.6	0.7	0.8	1
Viscosity (cP)	5.74	6.40	7.82	8.9	10.1	11.2	13.2	14.7	17.2	21.0

**Table 4 micromachines-13-00982-t004:** Viscosity of B98-benzyl alcohol solutions.

[B-98] (g/dL)	0	0.25	0.5	1	1.2	1.5	1.75	2	2.5	3
Viscosity (cP)	5.74	6.9	7.2	10.8	13.7	15.1	16.5	18.5	28.8	32.8

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
