# Peer review of "The Thickness and Structure of Dip-Coated Polymer Films in the Liquid and Solid States"

_micromachines, 2022, doi:10.3390/mi13070982_

Round 1

Reviewer 1 Report

The authors report on an interesting study regarding the liquid and solid state of dip coated polymer thin films. The experiments are well described and discussed. The obtained results fit well to LLD theory and are discussed in detail. I recommend only a few minor revisions.

1. All experiments and results are well documented. The only thing that I think can be improved a bit is the introduction/conclusion. The importance of the results is not immediately clear. The authors should try to highlight a bit more the results with regard to practical applications. In addition a short overview of polymer thin film deposition techniques would be nice in the introduction and why they are important. Maybe it's also good to point out the advantages of wet chemical thin film deposition techniques a bit more (e.g. lower costs), because polymer thin films can also be deposited via solvent-free CVD polymerization from the vapor phase (parylene, iCVD, MLD,...).

2. The figures could be a bit improved. In some figures are e.g. different fonts and font sizes.

Author Response

Question: All experiments and results are well documented. The only thing that I think can be improved a bit is the introduction/conclusion. The importance of the results is not immediately clear. The authors should try to highlight a bit more the results with regard to practical applications. In addition a short overview of polymer thin film deposition techniques would be nice in the introduction and why they are important.

Maybe it's also good to point out the advantages of wet chemical thin film deposition techniques a bit more (e.g. lower costs), because polymer thin films can also be deposited via solvent-free CVD polymerization from the vapor phase (parylene, iCVD, MLD,...).
I will list a few advantages of dip coating.
Answer: We thank the Reviewer for suggestions. The Introduction part was rewritten to include all these suggestions.
Question: The figures could be a bit improved. In some figures are e.g. different fonts and font sizes.
I will improve the figure quality.
Answer: We corrected all the figures as requested.

Reviewer 2 Report

The authors have studied the effects of polymer additives on the thickness of coated films in their liquid and solid states. This work provided new insights into understanding and helping in the fabrication of smooth polymeric films. In conclusion, this paper is well written and is recommended for publication after the following revisions:

1.      More up-to-date references with clear research gap should be given in the introduction part.

   2.      The main objective of the work must be written clearly and should briefly quantify the main research findings at the end of introduction section.

    3.      Conclusion must be written in more concisely with some perspective related to the future research work while quantifying main research findings.

     4.      English language should be carefully checked and some figures appear to be in poor format. Improve figure and graph quality if possible.

   5.      If possible, scanning electron microscopy images of the films with different polymers to observe the morphological changes more clearly.

6.      All graphs should follow the same format.

Author Response

Question: More up-to-date references with clear research gap should be given in the introduction part.
Answer: The Introduction part was rewritten and new references were included.
Question: The main objective of the work must be written clearly and should briefly quantify the main research findings at the end of introduction section.
Answer: The Introduction part was rewritten to include all these suggestions.
Question: Conclusion must be written in more concisely with some perspective related to the future research work while quantifying main research findings.
Answer: We edited Conclusion to address this suggestion.

Question: English language should be carefully checked and some figures appear to be in poor format. Improve figure and graph quality if possible.
Answer: We corrected the text and figures.

Question: We If possible, scanning electron microscopy images of the films with different polymers to observe the morphological changes more clearly.
Answer: We do not have these images
Question: We All graphs should follow the same format.
Answer: All figures were corrected.